# A Procedure for Constructing the Solution of a Nonlinear Fredholm Integro-Differential Equation of Second Order

**Rogério Martins Saldanha da Gama \***  **and Rogério Pazetto Saldanha da Gama**

Mechanical Engineering Department, Rio de Janeiro State University, São Fco Xavier Street 524, Rio de Janeiro 20550-013, Brazil
\* Correspondence: rogerio.gama@uerj.br

**Abstract:** In this work, a large class of integro-differential equations, arising from the description of heat transfer problems, is considered, particularly the nonlinear equations. We propose a procedure for constructing their solution in a very simple and reliable way in which the only needed tool is the same one employed to solve a linear second-order ordinary differential equation, subject to Robin boundary conditions. Proofs of the convergence, existence, and uniqueness are presented. Some special cases are simulated to illustrate the proposed tools.

**Keywords:** integro-differential equation; solution construction; application to heat transfer problems

**MSC:** 34B99; 80A21; 80A05

## 1. Introduction

In this work, the solution of a Fredholm integro-differential equation is represented as the limit of a sequence whose elements can be obtained from the minimization of quadratic functionals.

The integro-differential problem to be considered here has, as a particular case, the mathematical description of the heat transfer process in symmetrical sets of fins subjected to thermal radiation heat exchange.

In fact, any heat transfer problem involving nonconvex fins (or symmetrical sets of fins) at high temperature levels needs to consider thermal radiant heat transfer from/to the fin. The amount of reflected or emitted thermal radiation from a fin, directly reaching this same fin (or the same set of fins), is represented by an integral operator in the governing equation, giving rise to a second order integro-differential equation. The problems involving thermal radiation heat transfer are inherently nonlinear.

The procedure to be employed for constructing the exact solution may be used for obtaining approximations, for instance, by means of a finite difference scheme or by means of a finite element approximation (taking advantage of the quadratic functional).

Due to its applications in several areas of physics, mathematics, and engineering, Fredholm integro-differential equations continue to be an area of interest.

In the last two decades, their numerical simulations and mathematical analysis have been found with great frequency in scientific articles. Several procedures for solving integro-differential equations have been used, for instance, the Taylor polynomial approach [1,2], block-pulse functions [2,3], the CAS wavelet operational matrix [4–6], the Tau method [7], the Spectral Homotopy Analysis method [8,9], the Legendre collocation method [10], the Chebyshev finite difference method [11], the Decomposition Method [12], the Pade approximant [13], and other procedures [14,15].

The main contribution of this work lies in the extreme simplicity of the proposed procedure. It is presented a simple and reliable way to construct the exact solution for a given class of nonlinear Fredholm integro-differential equations, subject to Robin boundary conditions. The mentioned procedure can be also used to carry out numerical simulations

for the considered equations. The required tools are available to any undergraduate engineering student.

## 2. The Considered Problem

The main subject of this work is the problem represented in Equation (1), which generalizes the mathematical description of nonconvex sets of cylindrical fins [16–21], in which there is direct thermal radiant interchange among points (far positioned) of these fins. The main objective is to find the function $u$, the solution of

$$
\begin{aligned}
\frac{d^2u}{dx^2} - A\left\{\hat{f}(u)\right\} - B\left\{\hat{h}(u) - \int_0^1 \hat{h}(u)K(x,\xi)d\xi\right\} + C = 0 \quad & 0 < x < 1 \\
\frac{du}{dx} = \gamma_L(u - u_L), \quad & x = 0 \\
-\frac{du}{dx} = \gamma_R(u - u_R), \quad & x = 1
\end{aligned} \tag{1}
$$

in which $\gamma_L$ and $\gamma_R$ are positive constants, while $u_L$ and $u_R$ are nonnegative constants. In addition,

$$
\begin{aligned}
f = \hat{f}(u) &\to \text{nondecreasing function of } u, \text{ with } \hat{f}(0) \leq 0 \\
h = \hat{h}(u) &\to \text{nondecreasing function of } u, \text{ with } \hat{h}(0) = 0 \\
A = \widetilde{A}(x) &\geq 0 \to \text{known function} \\
B = \widetilde{B}(x) &> 0 \to \text{known function} \\
C = \widetilde{C}(x) &\geq 0 \to \text{known function} \\
K(x,\xi) &\geq 0, \quad 0 \leq x \leq 1, \quad 0 \leq \xi \leq 1 \\
0 \leq \int_0^1 K(x,\xi)d\xi &\leq \mu < 1, \quad 0 \leq x \leq 1, \quad \mu \to \text{constant}
\end{aligned} \tag{2}
$$

by means of a sequence whose elements can be easily obtained.

For instance, when problem (1) represents the heat transfer process in a set of two parallels fins, the kernel is given by [22].

$$
K(x,\xi) = \frac{(d/H)^2}{4\left((x-\xi)^2 + (d/H)^2\right)^{3/2}}, \tag{3}
$$

in which $d$ is the distance between the fins, and $H$ is the length of each fin.

When problem (1) represents the heat transfer process in a set of two fins (with an angle $2\theta$), the kernel is given by [22].

$$
K(x,\xi) = \frac{x\xi \sin^2(2\theta)}{4\left((4x\xi\sin\theta)^2 + (x-\xi)^2\right)^{3/2}}. \tag{4}
$$

When the surfaces are assumed to be black, the function $h = \hat{h}(u)$ is given by [23–26].

$$
h = \hat{h}(u) = B|T|^3 T, \quad B = \text{constant.} \tag{5}
$$

For a porous fin, the function $f = \hat{f}(u)$ is usually given by [20].

$$
f = \hat{f}(u) = A|T - T_\infty|(T - T_\infty), \quad A = \text{constant}, \quad T_\infty = \text{constant.} \tag{6}
$$

For a solid cylindrical fin, we usually have [25,26].

$$
f = \hat{f}(u) = A(T - T_\infty), \quad A = \text{constant}, \quad T_\infty = \text{constant.} \tag{7}
$$

The function $C = \widetilde{C}(x)$ plays the role of an external source. Many times, it is assumed to be zero everywhere [27].

### 3. The Solution $u$ Is Nonnegative

Aiming to prove that the solution of (1), denoted here by $u$, is nonnegative everywhere, let us begin by assuming that $u$ has a minimum within the interval $(0, 1)$. Denoting by $\overline{x}$ the point at which the minimum is reached, we have in a small vicinity [28,29] of this point that

$$-A\Big\{\hat{f}(u)\Big\} - B\Big\{\hat{h}(u) - \int_0^1 \hat{h}(u)K(x,\xi)d\xi\Big\} + C \le 0, \quad \overline{x} - \delta_1 < x < \overline{x} + \delta_2. \quad (8)$$

Taking into account (2), we may write

$$A\Big\{\hat{f}(u)\Big\} + B\Big\{\hat{h}(u) - \int_0^1 \hat{h}(u)K(x,\xi)d\xi\Big\} \ge 0, \quad \overline{x} - \delta_1 < x < \overline{x} + \delta_2. \quad (9)$$

Therefore, denoting by $u_{MIN}$ the minimum value of $u$, we have (within the considered vicinity)

$$A\Big\{\hat{f}(u_{MIN})\Big\} + B\Big\{\hat{h}(u_{MIN})\Big\} \ge 0 \quad \Rightarrow \quad u_{MIN} \ge 0. \quad (10)$$

On the other hand, let us assume that $u$ does not assume a minimum for $x \in (0, 1)$. In this case, we must have the minimum at $x = 0$ or at $x = 1$. If the minimum is reached at $x = 0$, then the derivative of $u$ is nonnegative at $x = 0$; hence,

$$\gamma_L(u_{MIN} - u_L) \ge 0 \quad \text{at} \quad x = 0 \quad \Rightarrow \quad u_{MIN} \ge 0, \quad (11)$$

and if the minimum is reached at $x = 1$, then the derivative of $u$ is nonpositive at $x = 1$; thus,

$$\gamma_R(u_{MIN} - u_R) \ge 0 \quad \text{at} \quad x = 1 \quad \Rightarrow \quad u_{MIN} \ge 0. \quad (12)$$

Hence, it is ensured that the solution of (1), denoted by $u$, is nonnegative everywhere.

### 4. An Upper Bound for the Solution $u$

Let us assume that $u$ assumes its maximum at the (interior) point $x = \overline{\overline{x}}$. In a sufficiently small neighborhood of this point, we have

$$-A\Big\{\hat{f}(u)\Big\} - B\Big\{\hat{h}(u) - \int_0^L \hat{h}(u)K(x,\xi)d\xi\Big\} + C \ge 0, \quad \overline{\overline{x}} - \delta_1 < x < \overline{\overline{x}} + \delta_2. \quad (13)$$

Hence,

$$\Big\{\max_{0 \le x \le 1} C\Big\} \ge \Big\{\min_{0 \le x \le 1} A\Big\}\Big\{\hat{f}(u_{MAX})\Big\} + \Big\{\min_{0 \le x \le 1} B\Big\}(1 - \mu)\Big\{\hat{h}(u_{MAX})\Big\}. \quad (14)$$

The above inequality consists of an upper bound for $u$, provided it assumes a maximum for $x \in (0, 1)$.

On the other hand, let us assume that $u$ does not reach a maximum for $x \in (0, 1)$. In this case, we must have the maximum at $x = 0$ or at $x = 1$. If the maximum is reached at $x = 0$, then the derivative of $u$ is nonpositive at $x = 0$; so,

$$\gamma_L(u_{MAX} - u_L) \le 0 \quad \text{at} \quad x = 0 \quad \Rightarrow \quad u_{MAX} \le u_L, \quad (15)$$

and if the maximum is reached at $x = 1$, then the derivative of $u$ is nonnegative at $x = 1$; hence,

$$\gamma_R(u_{MAX} - u_R) \le 0 \quad \text{at} \quad x = 1 \quad \Rightarrow \quad u_{MAX} \le u_R. \quad (16)$$

Therefore, we are able to evaluate an upper bound for the solution of (1), denoted by $u$. This value will be the larger one among $u_L$, $u_R$, and $u_*$, in which $u_*$ is the (unique) root of the equation below.

$$\left\{\max_{0\leq x\leq 1} C\right\} = \left\{\min_{0\leq x\leq 1} A\right\}\left\{\hat{f}(u_*)\right\} + \left\{\min_{0\leq x\leq 1} B\right\}(1-\mu)\left\{\hat{h}(u_*)\right\}. \tag{17}$$

## 5. Solution Construction

The solution of problem (1), denoted here by $u$, is given by the limit of the sequence $\left[\Psi^0, \Psi^1, \Psi^2, \Psi^3, \ldots\right]$, whose elements are given by

$$\beta^i = \alpha\Psi^i - A\,\hat{f}\left(\Psi^i\right) - B\left\{\hat{h}\left(\Psi^i\right) - \int_0^1 \hat{h}\left(\Psi^i\right) K(x,\xi)d\xi\right\} + C \quad \begin{array}{c} \frac{d^2\Psi^{i+1}}{dx^2} - \alpha\Psi^{i+1} + \beta^i = 0, \quad 0 < x < 1 \\ \\ \frac{d\Psi^{i+1}}{dx} = \gamma_L\left(\Psi^{i+1} - u_L\right), \quad x = 0 \\ \\ -\frac{d\Psi^{i+1}}{dx} = \gamma_R\left(\Psi^{i+1} - u_R\right), \quad x = 1 \end{array}, \tag{18}$$

in which $\Psi^0 \equiv 0$ and $\alpha$ is a (large) positive constant. In other words,

$$u \equiv \lim_{i\to\infty} \Psi^i. \tag{19}$$

## 6. The Behavior of the Sequence and the Constant $\alpha$

In order to show that $\left[\Psi^0, \Psi^1, \Psi^2, \Psi^3, \ldots\right]$ is a non-decreasing sequence, the first step is to show that $\Psi^1$ is nonnegative everywhere. For this, let us consider $i = 0$ and write

$$\beta^0 = \alpha\Psi^0 - A\,\hat{f}(\Psi^0) - B\left\{\hat{h}(\Psi^0) - \int_0^1 \hat{h}(\Psi^0) K(x,\xi)d\xi\right\} + C = -A\,\hat{f}(0) + \widetilde{C}(x) \geq 0 \quad \begin{array}{c} \frac{d^2\Psi^1}{dx^2} - \alpha\Psi^1 + \beta^0 = 0, \quad 0 < x < 1 \\ \\ \frac{d\Psi^1}{dx} = \gamma_L\left(\Psi^1 - u_L\right), \quad x = 0 \\ \\ -\frac{d\Psi^1}{dx} = \gamma_R\left(\Psi^1 - u_R\right), \quad x = 1 \end{array}. \tag{20}$$

Suppose that $\Psi^1$ assumes a minimum for $x \in (0,1)$. In this case, within a sufficiently small vicinity of the point $x = \bar{x}$, in which $\Psi^1$ reaches its minimum, we must have (denoting the minimum by $\Psi^1_{MIN}$).

$$-\alpha\Psi^1_{MIN} + \beta^0 \leq 0 \quad \Rightarrow \quad \Psi^1_{MIN} \geq \frac{\beta^0}{\alpha} \geq 0. \tag{21}$$

On the other hand, if $\Psi^1$ assumes its minimum at $x = 0$, we must have

$$\gamma_L\left(\Psi^1_{MIN} - u_L\right) \geq 0 \quad \Rightarrow \quad \Psi^1_{MIN} \geq u_L \geq 0, \tag{22}$$

while, if the minimum is reached at $x = 1$, we have

$$\gamma_R\left(\Psi^1_{MIN} - u_R\right) \geq 0 \quad \Rightarrow \quad \Psi^1_{MIN} \geq u_R \geq 0. \tag{23}$$

Therefore, the minimum of $\Psi^1$ is nonnegative. In other words, the function $\hat{\Psi}^1(x)$ is nonnegative everywhere. In addition, we have proven that $\Psi^1 \geq \Psi^0 \equiv 0$.

Now, let us consider (18) for two consecutive values of $i$. The difference yields

$$\frac{d^2\left(\Psi^{i+1}-\Psi^i\right)}{dx^2} - \alpha\left(\Psi^{i+1}-\Psi^i\right) + \left(\beta^i-\beta^{i-1}\right) = 0, \quad 0 < x < 1$$
$$\beta^i - \beta^{i-1} = \alpha\left(\Psi^i-\Psi^{i-1}\right) - A\left\{\hat{f}\left(\Psi^i\right)-\hat{f}\left(\Psi^{i-1}\right)\right\} -$$
$$- B\left\{\hat{h}\left(\Psi^i\right)-\hat{h}\left(\Psi^{i-1}\right) - \int_0^1\left(\hat{h}\left(\Psi^i\right)-\hat{h}\left(\Psi^{i-1}\right)\right)K(x,\xi)d\xi\right\}. \qquad (24)$$
$$\frac{d\left(\Psi^{i+1}-\Psi^i\right)}{dx} = \gamma_L\left(\Psi^{i+1}-\Psi^i\right), \quad x=0$$
$$-\frac{d\left(\Psi^{i+1}-\Psi^i\right)}{dx} = \gamma_R\left(\Psi^{i+1}-\Psi^i\right), \quad x=1$$

If the difference $\Psi^{i+1}-\Psi^i$ assumes its minimum at the point $\overline{x} \in (0,1)$, then, in a sufficiently small neighborhood of this point, we must have

$$\alpha\left(\Psi^{i+1}-\Psi^i\right) \geq \left(\beta^i-\beta^{i-1}\right). \qquad (25)$$

If the minimum is reached at $x=0$, we must have

$$\gamma_L\left(\Psi^{i+1}-\Psi^i\right) \geq 0 \quad \Rightarrow \quad \Psi^{i+1}-\Psi^i \geq 0, \qquad (26)$$

while, if the minimum is reached at $x=1$, we have

$$\gamma_R\left(\Psi^{i+1}-\Psi^i\right) \geq 0 \quad \Rightarrow \quad \Psi^{i+1}-\Psi^i \geq 0. \qquad (27)$$

In this way, in order to ensure that $\Psi^{i+1}-\Psi^i \geq 0$ everywhere, we must ensure that $\beta^i-\beta^{i-1} \geq 0$ for all $x \in (0,1)$.

This condition is always satisfied when the constant $\alpha$ is chosen in such a way that

$$\alpha\left(\Psi^i-\Psi^{i-1}\right) \geq$$
$$\geq A\left\{\hat{f}\left(\Psi^i\right)-\hat{f}\left(\Psi^{i-1}\right)\right\} + B\left\{\hat{h}\left(\Psi^i\right)-\hat{h}\left(\Psi^{i-1}\right) - \int_0^1\left(\hat{h}\left(\Psi^i\right)-\hat{h}\left(\Psi^{i-1}\right)\right)K(x,\xi)d\xi\right\} \qquad (28)$$

for all $x \in (0,1)$.

Hence, if $\Psi^i \geq \Psi^{i-1}$, a sufficient (not necessary) condition for ensuring that $\Psi^{i+1} \geq \Psi^i$ is the following

$$\alpha \geq \frac{A\left\{\hat{f}\left(\Psi^i\right)-\hat{f}\left(\Psi^{i-1}\right)\right\} + B\left\{\hat{h}\left(\Psi^i\right)-\hat{h}\left(\Psi^{i-1}\right)\right\}}{\Psi^i-\Psi^{i-1}}, \quad x \in (0,1), \quad i=1,2,3,\dots \quad (29)$$

Therefore, since $\Psi^1 \geq \Psi^0$, we ensure that $\Psi^2 \geq \Psi^1$ provided

$$\alpha \geq \frac{A\left\{\hat{f}\left(\Psi^1\right)-\hat{f}\left(\Psi^0\right)\right\} - B\left\{\hat{h}\left(\Psi^1\right)-\hat{h}\left(\Psi^0\right)\right\}}{\Psi^1-\Psi^0}. \qquad (30)$$

Repeating this procedure we have, for sufficiently large $\alpha$, that $\Psi^{i+1} \geq \Psi^i$.

## 7. The Solution $u$ as an Upper Bound for $\Psi^i$

From (1) and (18), we may write

$$\frac{d^2\left(u-\Psi^{i+1}\right)}{dx^2} + \alpha\left(\Psi^{i+1}-\Psi^i\right) -$$
$$- A\left\{\hat{f}(u)-\hat{f}\left(\Psi^i\right)\right\} - B\left\{\hat{h}(u)-\hat{h}\left(\Psi^i\right) - \int_0^1\left(\hat{h}(u)-\hat{h}\left(\Psi^i\right)\right)K(x,\xi)d\xi\right\} = 0 \qquad (31)$$
$$\frac{d\left(u-\Psi^{i+1}\right)}{dx} = \gamma_L\left(u-\Psi^{i+1}\right), \quad x=0$$
$$-\frac{d\left(u-\Psi^{i+1}\right)}{dx} = \gamma_R\left(u-\Psi^{i+1}\right), \quad x=1$$

Let us consider that $u - \Psi^{i+1}$ assumes its minimum at the interior point $x = \overline{x}$. So, in a sufficiently small neighborhood of this point, we must have

$$\alpha\left(\Psi^{i+1} - \Psi^i\right) - A\left\{\hat{f}(u) - \hat{f}\left(\Psi^i\right)\right\} - B\left\{\hat{h}(u) - \hat{h}\left(\Psi^i\right) - \int_0^1 \left(\hat{h}(u) - \hat{h}\left(\Psi^i\right)\right)K(x,\xi)d\xi\right\} \le 0 \tag{32}$$

or, in a more convenient form, assuming that (29) holds,

$$\begin{aligned} \alpha\left(\Psi^{i+1} - \Psi^i\right) - A\left\{\hat{f}\left(\Psi^{i+1}\right) - \hat{f}\left(\Psi^i\right)\right\} - \\ -B\left\{\hat{h}\left(\Psi^{i+1}\right) - \hat{h}\left(\Psi^i\right) - \int_0^1 \left(\hat{h}(u) - \hat{h}\left(\Psi^i\right)\right)K(x,\xi)d\xi\right\} \le \\ \le A\left\{\hat{f}(u) - \hat{f}\left(\Psi^{i+1}\right)\right\} + B\left\{\hat{h}(u) - \hat{h}\left(\Psi^{i+1}\right)\right\} \end{aligned} \tag{33}$$

Since $u$ is nonnegative everywhere, and $\Psi^0 \equiv 0$ (and taking into account (29)—the definition of $\alpha$), we conclude that

$$A\left\{\hat{f}(u) - \hat{f}\left(\Psi^1\right)\right\} + B\left\{\hat{h}(u) - \hat{h}\left(\Psi^1\right)\right\} \ge 0 \tag{34}$$

in the neighborhood of $x = \overline{x}$. Therefore, in this case, $u \ge \Psi^1$ everywhere.

On the other hand, if the difference $u - \Psi^{i+1}$ assumes its minimum at $x = 0$, we must have

$$\gamma_L\left(u - \Psi^{i+1}\right) \ge 0 \quad \Rightarrow \quad u - \Psi^{i+1} \ge 0, \text{ at } x = 0, \tag{35}$$

while, if the minimum is reached at $x = 1$, we have

$$\gamma_R\left(u - \Psi^{i+1}\right) \ge 0 \quad \Rightarrow \quad u - \Psi^{i+1} \ge 0, \text{ at } x = 1. \tag{36}$$

Repeating this procedure, we can conclude that the minimum of $u - \Psi^{i+1}$ is nonnegative. Therefore,

$$u \ge \ldots \ge \Psi^{i+1} \ge \Psi^i \ge \ldots \ge \Psi^3 \ge \Psi^2 \ge \Psi^1 \ge \Psi^0 \equiv 0, \text{ for } x \in [0,1]. \tag{37}$$

In other words, the solution of the original problem represents an upper bound for the sequence $\left[\Psi^0, \Psi^1, \Psi^2, \Psi^3, \ldots\right]$. This fact ensures the convergence.

Since the solution $u$ is nonnegative and has a known upper bound, the constant $\alpha$ may be chosen from the following formula (this is not a necessary choice)

$$\alpha = \left\{\max_{0 \le x \le 1} A\right\}\left\{\max_{0 \le \theta \le u_{MAX}} \frac{df}{d\theta}\right\} + \left\{\max_{0 \le x \le 1} B\right\}\left\{\max_{0 \le \theta \le u_{MAX}} \frac{dh}{d\theta}\right\}, \tag{38}$$

provided the derivatives of $f$ and of $h$ are bounded.

It is remarkable that the convergence may be reached for lower values of the constant $\alpha$.

## 8. Solution Uniqueness

In order to demonstrate that the limit of the sequence $\left[\Psi^0, \Psi^1, \Psi^2, \Psi^3, \ldots\right]$ is the unique solution of problem (1), let us assume that $u$ is different from $\Psi^\infty$ (the limit).

Since (37) holds, we must have

$$u \ge \Psi^\infty, \quad 0 \le x \le 1, \tag{39}$$

and we only need to show that the maximum of $u - \Psi^\infty$ is not positive.

For this, let us assume that $u - \Psi^\infty$ assumes its maximum at the point $\overline{\overline{x}} \in (0,1)$. So, in a sufficiently small vicinity of this point, we must have

$$-A\left\{\hat{f}(u) - \hat{f}(\Psi^\infty)\right\} - B\left\{\hat{h}(u) - \hat{h}(\Psi^\infty) - \int_0^1 \left(\hat{h}(u) - \hat{h}(\Psi^\infty)\right)K(x,\xi)d\xi\right\} \ge 0. \tag{40}$$

So, from the definition of the constant $\alpha$, we have, in the considered vicinity,

$$\alpha(u - \Psi^\infty) \le 0 \quad \Rightarrow \quad u \le \Psi^\infty. \tag{41}$$

Now, suppose that $u - \Psi^\infty$ assumes its maximum at the point $x = 0$. In this case, we have

$$\gamma_L(u - \Psi^\infty) \le 0 \quad \Rightarrow \quad u \le \Psi^\infty, \text{ at } x = 0. \tag{42}$$

On the other hand, if $u - \Psi^\infty$ assumes its maximum at the point $x = 1$, we have

$$\gamma_R(u - \Psi^\infty) \le 0 \quad \Rightarrow \quad u \le \Psi^\infty, \text{ at } x = 1. \tag{43}$$

Consequently, it is demonstrated that the solution is unique and is represented by the limit of the sequence $\left[\Psi^0, \Psi^1, \Psi^2, \Psi^3, \ldots\right]$.

## 9. Variational Formulation

The solution of problem (18), denoted by $\Psi^{i+1}$, is the function that minimizes the (quadratic) functional below

$$I[v] = \frac{1}{2}\int_0^1 \left\{ \left(\frac{dv}{dx}\right)^2 + \alpha v^2 \right\} dx - \int_0^1 \beta^i v \, dx + \frac{1}{2}\gamma_L \left[(v - u_L)^2\right]_{x=0} + \frac{1}{2}\gamma_R \left[(v - u_R)^2\right]_{x=1}. \tag{44}$$

The first variation of $I[v]$ is given by [30]

$$\delta I[v] = \int_0^1 \left\{ \left(\frac{dv}{dx}\right)\frac{d\delta v}{dx} + \alpha v \delta v \right\} dx - \int_0^1 \beta^i \delta v \, dx + \gamma_L[(v - u_L)\delta v]_{x=0} + \gamma_R[(v - u_R)\delta v]_{x=1}, \tag{45}$$

or in a more convenient form by

$$\delta I[v] = \int_0^1 \left\{ \frac{d}{dx}\left(\left(\frac{dv}{dx}\right)\delta v\right) - \frac{d^2 v}{dx^2}\delta v + \alpha v \delta v \right\} dx - \int_0^1 \beta^i \delta v \, dx + \gamma_L[(v - u_L)\delta v]_{x=0} + \gamma_R[(v - u_R)\delta v]_{x=1} \tag{46}$$

in which $\delta v$ is any admissible variation.

Considering that

$$\int_0^1 \frac{d}{dx}\left(\left(\frac{dv}{dx}\right)\delta v\right) dx = \left[\left(\frac{dv}{dx}\right)\delta v\right]_{x=1} - \left[\left(\frac{dv}{dx}\right)\delta v\right]_{x=0}, \tag{47}$$

we may write

$$\delta I[v] = \int_0^1 \left\{ -\frac{d^2 v}{dx^2}\delta v + \alpha v \delta v \right\} dx - \int_0^1 \beta^i \delta v \, dx + \gamma_L[(v - u_L)\delta v]_{x=0} + \\ + \gamma_R[(v - u_R)\delta v]_{x=1} + \left[\left(\frac{dv}{dx}\right)\delta v\right]_{x=1} - \left[\left(\frac{dv}{dx}\right)\delta v\right]_{x=0}. \tag{48}$$

The extremum of the functional is obtained making $\delta I[v] = 0$. Considering that $\delta v$ is arbitrary, we obtain the Euler–Lagrange equation and the natural boundary conditions as follows

$$\begin{array}{ll} -\frac{d^2 v}{dx^2} + \alpha v - \beta^i = 0, & 0 < x < 1 \\ \gamma_L(v - u_L) - \left(\frac{dv}{dx}\right) = 0, & x = 0 \\ \gamma_R(v - u_R) + \left(\frac{dv}{dx}\right) = 0, & x = 1 \end{array}. \tag{49}$$

Problems (44) and (18) are the same problem, since $\beta^i$ is known.

Since the functional $I[v]$ is strictly convex (see Appendix A), its extremum is unique and corresponds to a minimum [29,30]. Thus, each element $\Psi^{i+1}$ may be obtained from the minimization of $I[v]$.

The existence of this minimum is ensured, since the functional is coercive [31].

## 10. A Numerical Approximation

The procedure proposed for constructing the solution of (1) may be used for obtaining numerical approximations too.

To illustrate this fact, let us consider the following (piecewise linear) approximation for the element $\Psi^i$, given by

$$\Psi^i = \left(\Psi^i_{j+1} - \Psi^i_j\right)\left(\frac{x - x_j}{\Delta x}\right) + \Psi^i_j, \; x_j \le x \le x_{j+1}, \; x_j = (j-1)\Delta x, \; j = 1, 2, \ldots, N, \; \Delta x = \frac{1}{N} \tag{50}$$

in which $\Psi^i_j$ represents the approximation for $\Psi^i$ at the spatial point $x_j$, and $\Delta x = 1/N$.

In this case, the functional $I[v]$ becomes, for each $i$, the following function

$$F^{i+1}(v_1, v_2, \ldots, v_N, v_{N+1}) = \tfrac{1}{2}\gamma_L(v_1 - u_L)^2 + \tfrac{1}{2}\gamma_R(v_N - u_R)^2 +$$
$$+\tfrac{1}{2}\sum_{j=1}^{N}\int_{x_j}^{x_{j+1}}\left\{\left(\frac{v_{j+1}-v_j}{\Delta x}\right)^2 + \alpha\left(\left(v_{j+1} - v_j\right)\left(\frac{x-x_j}{\Delta x}\right) + v_j\right)^2 - 2\beta^i\left(\left(v_{j+1} - v_j\right)\left(\frac{x-x_j}{\Delta x}\right) + v_j\right)\right\}dx \tag{51}$$

in which $\beta^i$ is given by (considering the piecewise approximation (50))

$$\beta^i = \alpha\left(\left(\Psi^i_{j+1} - \Psi^i_j\right)\left(\frac{x-x_j}{\Delta x}\right) + \Psi^i_j\right) - A\,\hat{f}\left(\left(\Psi^i_{j+1} - \Psi^i_j\right)\left(\frac{x-x_j}{\Delta x}\right) + \Psi^i_j\right) -$$
$$-B\left\{\hat{h}\left(\left(\Psi^i_{j+1} - \Psi^i_j\right)\left(\frac{x-x_j}{\Delta x}\right) + \Psi^i_j\right) - \int_0^1 \hat{h}\left(\left(\Psi^i_{j+1} - \Psi^i_j\right)\left(\frac{\xi-x_j}{\Delta x}\right) + \Psi^i_j\right)K(x,\xi)d\xi\right\} + C \tag{52}$$

The values of $v_j$ that minimize the function $F^{i+1}(v_1, v_2, \ldots, v_N, v_{N+1})$ defined in (46) are exactly the values of $\Psi^{i+1}_j$, obtained from the following system (linear),

$$\frac{\partial}{\partial v_j}\left\{F^{i+1}(v_1, v_2, \ldots, v_N, v_{N+1})\right\} = 0, \quad j = 1, 2, \ldots, N+1. \tag{53}$$

## 11. Two Examples (With Known Exact Solutions)

Let us consider the following problem

$$\begin{aligned}\frac{d^2u}{dx^2} - u - \left\{|u|^3u - \int_0^1 |u|^3u\,\tfrac{5}{31}d\xi\right\} + x + (1+x)^4 &= 0, \quad 0 < x < 1 \\ \frac{du}{dx} &= u, \quad x = 0 \\ -\frac{du}{dx} &= u - 3, \quad x = 1\end{aligned} \tag{54}$$

whose exact solution is given by

$$u = 1 + x, \quad 0 \le x \le 1. \tag{55}$$

Comparing (53) with (1) we have that

$$\begin{aligned}A = B = 1, \quad C = x + (1+x)^4, \quad \gamma_L = \gamma_R = 1, \quad u_L = 0, \quad u_R = 3 \\ K(x,\xi) = 5/31, \quad \hat{f}(u) = u, \quad \hat{h}(u) = |u|^3u\end{aligned} \tag{56}$$

Some elements of the sequence $\left[\Psi^0, \Psi^1, \Psi^2, \Psi^3, \ldots\right]$, approximated by (50), are shown in Figure 1, as well as the exact solution.

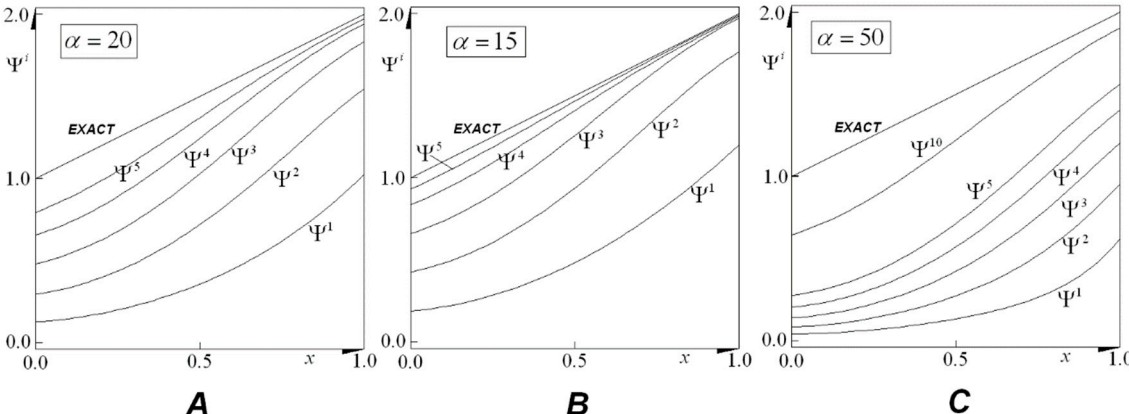

**Figure 1.** Some elements of the sequence $\left[\Psi^0, \Psi^1, \Psi^2, \Psi^3, \ldots\right]$ obtained with three different values of the constant $\alpha$ ($\alpha = 20$ (**A**), $\alpha = 15$ (**B**), $\alpha = 50$ (**C**)). In all cases, $N = 50$.

It is to be noticed that the speed of convergence was strongly affected by the value of $\alpha$. As $\alpha$ increased, the speed of convergence decreased. Nevertheless, $\alpha$ cannot be very small, since this may give rise to nonincreasing sequences, and the convergence may not be achieved.

To illustrate the role of the constant $\alpha$ in an explicit way, let us consider the following problem:

$$
\begin{array}{c}
\frac{d^2u}{dx^2} - \frac{1}{2}u - \left\{ |u|^3u - \int_0^1 |u|^3u\,\frac{1}{2}d\xi \right\} + 1 = 0, \quad 0 < x < 1 \\
\frac{du}{dx} = u - 1, \quad x = 0 \\
-\frac{du}{dx} = u - 1, \quad x = 1
\end{array}
\tag{57}
$$

whose exact solution is given by

$$
u = 1, \quad 0 \le x \le 1.
\tag{58}
$$

The elements of the sequence are obtained from

$$
\begin{array}{c}
\frac{d^2\Psi^{i+1}}{dx^2} - \alpha\Psi^{i+1} + \beta^i = 0, \quad 0 < x < 1 \\
\beta^i = \alpha\Psi^i - \frac{1}{2}\Psi^i - \left\{ \left|\Psi^i\right|^3\Psi^i - \int_0^1 \left|\Psi^i\right|^3\Psi^i\,\frac{1}{2}d\xi \right\} + 1 \\
\frac{d\Psi^{i+1}}{dx} = \left(\Psi^{i+1} - 1\right), \quad x = 0 \\
-\frac{d\Psi^{i+1}}{dx} = \left(\Psi^{i+1} - 1\right), \quad x = 1
\end{array}
\tag{59}
$$

Let us assume that for each $i$, $\Psi^i$ is a constant for $0 \le x \le 1$. This is equivalent to imposing $\Psi^i_2 = \Psi^i_1$, assuming $N = 1$. In this case, the functional becomes the function

$$
F^{i+1}(v) = \frac{1}{2}(v-1)^2 + \frac{1}{2}(v-1)^2 + \frac{1}{2}\int_0^1 \left\{ \alpha(v)^2 - 2\beta^i(v) \right\}dx.
\tag{60}
$$

So, the constant value for $\Psi^{i+1}$ will be obtained from the minimization of the function $F^{i+1}(v)$. In other words, it will be the root of

$$
\frac{d}{dv}F^{i+1}(v) = (v-1) + (v-1) + \alpha v - \int_0^1 \beta^i dx.
\tag{61}
$$

The root of the above expression will be exactly the value of $\Psi^{i+1}$ (assumed a constant for each $i$), given by

$$\left(\Psi^{i+1} - 1\right) + \left(\Psi^{i+1} - 1\right) + \alpha\Psi^{i+1} - \int_0^1 \beta^i dx = 0 \quad \Rightarrow \quad \Psi^{i+1} = \frac{\int_0^1 \beta^i dx + 2}{2 + \alpha}, \tag{62}$$

or considering the expression for $\beta^i$, we have

$$\Psi^{i+1} = \frac{2 + \beta^i}{2 + \alpha} = \frac{3 + \left(\alpha - \frac{1}{2}\right)\Psi^i - \frac{1}{2}\left(\Psi^i\right)^4}{2 + \alpha}. \tag{63}$$

Table 1 illustrates the convergence for several values of the constant $\alpha$. Notice that for $\alpha = 0.1$ and for $\alpha = 0.01$, the convergence was not reached. For $\alpha = 2$, we reached convergence, but the sequence was not a nondecreasing one. The speed of convergence, for $\alpha \geq 5$, decreased as $\alpha$ increased.

In other words, each column of Table 1 represents the elements: $\Psi^1$, ..., $\Psi^{50}$, obtained with nine different values of the constant $\alpha$.

**Table 1.** The constant $\Psi^i$ obtained for nine different values of $\alpha$.

| $\alpha \rightarrow$ | 100 | 50 | 20 | 10 | 5 | 2 | 0.5 | 0.1 | 0.01 |
|---|---|---|---|---|---|---|---|---|---|
| i = 1 | 0.0294 | 0.0577 | 0.1364 | 0.2500 | 0.4286 | 0.7500 | 1.200 | 1.4286 | 1.4925 |
| i = 2 | 0.0581 | 0.1126 | 0.2572 | 0.4478 | 0.7017 | 0.9917 | 0.7853 | 0.1648 | −0.1058 |
| i = 3 | 0.0861 | 0.1649 | 0.3643 | 0.6028 | 0.8623 | 1.0010 | 1.1239 | 1.3970 | 1.5183 |
| i = 4 | 0.1134 | 0.2146 | 0.4588 | 0.7217 | 0.9434 | 0.9999 | 0.8808 | 0.2556 | −0.1995 |
| i = 5 | 0.1400 | 0.2620 | 0.5420 | 0.8101 | 0.9785 | 1.0000 | 1.0796 | 1.3789 | 1.5408 |
| i = 6 | 0.1660 | 0.3070 | 0.6149 | 0.8734 | 0.9921 | 1.0000 | 0.9283 | 0.3053 | −0.285 |
| i = 7 | 0.1913 | 0.3499 | 0.6781 | 0.9172 | 0.9972 | 1.0000 | 1.0515 | 1.3684 | 1.5604 |
| i = 8 | 0.2161 | 0.3906 | 0.7326 | 0.9466 | 0.9990 | 1.0000 | 0.9555 | 0.3332 | −0.3625 |
| i = 9 | 0.2402 | 0.4293 | 0.7792 | 0.9659 | 0.9996 | 1.0000 | 1.0333 | 1.3622 | 1.5766 |
| i = 10 | 0.2637 | 0.4660 | 0.8186 | 0.9784 | 0.9999 | 1.0000 | 0.9720 | 0.3494 | −0.4288 |
| i = 11 | 0.2866 | 0.5009 | 0.8517 | 0.9864 | 1.0000 | 1.0000 | 1.0215 | 1.3585 | 1.5887 |
| i = 12 | 0.3090 | 0.5339 | 0.8794 | 0.9915 | 1.0000 | 1.0000 | 0.9823 | 0.3589 | −0.4793 |
| i = 13 | 0.3307 | 0.5651 | 0.9022 | 0.9946 | 1.0000 | 1.0000 | 1.0138 | 1.3563 | 1.5963 |
| i = 14 | 0.3520 | 0.5947 | 0.9210 | 0.9966 | 1.0000 | 1.0000 | 0.9887 | 0.3646 | −0.5116 |
| i = 15 | 0.3727 | 0.6226 | 0.9363 | 0.9979 | 1.0000 | 1.0000 | 1.0089 | 1.3549 | 1.6002 |
| i = 16 | 0.3929 | 0.6489 | 0.9488 | 0.9987 | 1.0000 | 1.0000 | 0.9928 | 0.3681 | −0.5287 |
| i = 17 | 0.4125 | 0.6737 | 0.9590 | 0.9992 | 1.0000 | 1.0000 | 1.0057 | 1.3541 | 1.6020 |
| i = 18 | 0.4317 | 0.6970 | 0.9671 | 0.9995 | 1.0000 | 1.0000 | 0.9954 | 0.3702 | −0.5364 |
| i = 19 | 0.4504 | 0.7189 | 0.9737 | 0.9997 | 1.0000 | 1.0000 | 1.0036 | 1.3536 | 1.6027 |
| i = 20 | 0.4685 | 0.7395 | 0.9790 | 0.9998 | 1.0000 | 1.0000 | 0.9971 | 0.3715 | −0.5395 |
| i = 21 | 0.4862 | 0.7587 | 0.9832 | 0.9999 | 1.0000 | 1.0000 | 1.0023 | 1.3533 | 1.6030 |
| i = 22 | 0.5035 | 0.7768 | 0.9866 | 0.9999 | 1.0000 | 1.0000 | 0.9981 | 0.3723 | −0.5407 |
| i = 23 | 0.5202 | 0.7936 | 0.9893 | 1.0000 | 1.0000 | 1.0000 | 1.0015 | 1.3531 | 1.6031 |
| i = 24 | 0.5365 | 0.8093 | 0.9915 | 1.0000 | 1.0000 | 1.0000 | 0.9988 | 0.3727 | −0.5411 |
| i = 25 | 0.5524 | 0.8240 | 0.9932 | 1.0000 | 1.0000 | 1.0000 | 1.0010 | 1.3530 | 1.6031 |
| i = 26 | 0.5678 | 0.8376 | 0.9946 | 1.0000 | 1.0000 | 1.0000 | 0.9992 | 0.3730 | −0.5413 |
| i = 27 | 0.5828 | 0.8503 | 0.9957 | 1.0000 | 1.0000 | 1.0000 | 1.0006 | 1.3529 | 1.6031 |
| i = 28 | 0.5973 | 0.8621 | 0.9966 | 1.0000 | 1.0000 | 1.0000 | 0.9995 | 0.3732 | −0.5414 |
| i = 29 | 0.6115 | 0.8730 | 0.9973 | 1.0000 | 1.0000 | 1.0000 | 1.0004 | 1.3529 | 1.6031 |
| i = 30 | 0.6252 | 0.8832 | 0.9978 | 1.0000 | 1.0000 | 1.0000 | 0.9997 | 0.3733 | −0.5414 |
| i = 31 | 0.6386 | 0.8926 | 0.9983 | 1.0000 | 1.0000 | 1.0000 | 1.0003 | 1.3528 | 1.6031 |
| i = 32 | 0.6515 | 0.9012 | 0.9986 | 1.0000 | 1.0000 | 1.0000 | 0.9998 | 0.3734 | −0.5414 |
| i = 33 | 0.6641 | 0.9093 | 0.9989 | 1.0000 | 1.0000 | 1.0000 | 1.0002 | 1.3528 | 1.6031 |

**Table 1.** *Cont.*

| $\alpha \rightarrow$ | 100 | 50 | 20 | 10 | 5 | 2 | 0.5 | 0.1 | 0.01 |
|---|---|---|---|---|---|---|---|---|---|
| i = 34 | 0.6762 | 0.9167 | 0.9991 | 1.0000 | 1.0000 | 1.0000 | 0.9999 | 0.3734 | −0.5414 |
| i = 35 | 0.6881 | 0.9235 | 0.9993 | 1.0000 | 1.0000 | 1.0000 | 1.0001 | 1.3528 | 1.6031 |
| i = 36 | 0.6995 | 0.9298 | 0.9995 | 1.0000 | 1.0000 | 1.0000 | 0.9999 | 0.3734 | −0.5414 |
| i = 37 | 0.7106 | 0.9356 | 0.9996 | 1.0000 | 1.0000 | 1.0000 | 1.0001 | 1.3528 | 1.6031 |
| i = 38 | 0.7213 | 0.9409 | 0.9997 | 1.0000 | 1.0000 | 1.0000 | 0.9999 | 0.3735 | −0.5414 |
| i = 39 | 0.7318 | 0.9459 | 0.9997 | 1.0000 | 1.0000 | 1.0000 | 1.0000 | 1.3528 | 1.6031 |
| i = 40 | 0.7418 | 0.9504 | 0.9998 | 1.0000 | 1.0000 | 1.0000 | 1.0000 | 0.3735 | −0.5414 |
| i = 41 | 0.7516 | 0.9545 | 0.9998 | 1.0000 | 1.0000 | 1.0000 | 1.0000 | 1.3528 | 1.6031 |
| i = 42 | 0.7610 | 0.9584 | 0.9999 | 1.0000 | 1.0000 | 1.0000 | 1.0000 | 0.3735 | −0.5414 |
| i = 43 | 0.7701 | 0.9619 | 0.9999 | 1.0000 | 1.0000 | 1.0000 | 1.0000 | 1.3528 | 1.6031 |
| i = 44 | 0.7789 | 0.9651 | 0.9999 | 1.0000 | 1.0000 | 1.0000 | 1.0000 | 0.3735 | −0.5414 |
| i = 45 | 0.7874 | 0.9680 | 0.9999 | 1.0000 | 1.0000 | 1.0000 | 1.0000 | 1.3528 | 1.6031 |
| i = 46 | 0.7957 | 0.9707 | 0.9999 | 1.0000 | 1.0000 | 1.0000 | 1.0000 | 0.3735 | −0.5414 |
| i = 47 | 0.8036 | 0.9732 | 1.0000 | 1.0000 | 1.0000 | 1.0000 | 1.0000 | 1.3528 | 1.6031 |
| i = 48 | 0.8113 | 0.9755 | 1.0000 | 1.0000 | 1.0000 | 1.0000 | 1.0000 | 0.3735 | −0.5414 |
| i = 49 | 0.8187 | 0.9776 | 1.0000 | 1.0000 | 1.0000 | 1.0000 | 1.0000 | 1.3528 | 1.6031 |
| i = 50 | 0.8258 | 0.9795 | 1.0000 | 1.0000 | 1.0000 | 1.0000 | 1.0000 | 0.3735 | −0.5414 |

## 12. Conclusions

The procedure proposed here differs from others due to its simplicity. Only basic tools are needed, which are available for undergraduate students. This is the main novelty and contribution of this work.

In addition, since the exact solution is represented by the limit of a sequence, there is no limit of accuracy when employing a numerical approximation.

The proposed procedure may be used for problems involving Dirichlet and some Neumann boundary conditions. This may be performed by employing very large or very small values for $\gamma_L$ and $\gamma_R$. When $\gamma_L$ (and/or $\gamma_R$) is very large, we approximate a Dirichlet boundary condition. When $\gamma_L$ (and/or $\gamma_R$) is very small, we approximate a Neumann boundary condition (in this case representing an insulated edge).

When modeling a heat transfer problem involving high temperature levels, thermal radiant heat transfer plays a meaningful role, since the thermal interaction among far-positioned points becomes more significant as the temperature levels become larger. This fact gives rise to an integral operator, which represents the amount of thermal radiant energy arriving at each point of the body.

Besides the heat transfer phenomena, in which the thermal radiation plays a non-negligible role, integro-differential equations are found in several other branches, such as optimal control problems [32]. Integro-differential equations involving fractional derivatives also consist of a potential issue to be explored due to their increasing interest and applicability [33–39].

The heat transfer process in a nonsymmetrical system of fins, as well as in multiphase bodies [40], gives rise to systems of second-order integro-differential equations. It seems logical that, with some adjustments, the procedure proposed here may be extended to such a class of problems.

**Author Contributions:** Methodology, R.M.S.d.G.; Software, R.P.S.d.G.; Formal analysis, R.M.S.d.G.; Investigation, R.M.S.d.G.; Writing–original draft, R.P.S.d.G. All authors have read and agreed to the published version of the manuscript.

**Funding:** The author: R. M. S. Gama, gratefully acknowledges the support provided by Brazilian Agency CNPq (Grant 306364/2018-1) and by the Brazilian Agency CAPES (Finance code 001).

**Informed Consent Statement:** Not applicable.

**Conflicts of Interest:** The authors declare no conflict of interest.

## Appendix A. On the Convexity of the Functional $I[v]$

A functional $I[v]$ is said to be strictly convex, if and only if

$$\theta I[v_1] + (1-\theta)I[v_2] > I[\theta v_1 + (1-\theta)v_2], \quad \theta \in (0,1), \quad v_1 \neq v_2. \tag{A1}$$

So, the functional defined in (44) is said to be strictly convex, if and only if the following inequality holds

$$
\begin{aligned}
&\theta\left\{\frac{1}{2}\int_0^1\left\{\left(\frac{dv_1}{dx}\right)^2 + \alpha v_1^2\right\}dx - \int_0^1 \beta^i v_1 dx + \frac{1}{2}\gamma_L\left[(v_1-u_L)^2\right]_{x=0} + \frac{1}{2}\gamma_R\left[(v_1-u_R)^2\right]_{x=1}\right\} + \\
&+(1-\theta)\left\{\frac{1}{2}\int_0^1\left\{\left(\frac{dv_2}{dx}\right)^2 + \alpha v_2^2\right\}dx - \int_0^1 \beta^i v_2 dx + \frac{1}{2}\gamma_L\left[(v_2-u_L)^2\right]_{x=0} + \frac{1}{2}\gamma_R\left[(v_2-u_R)^2\right]_{x=1}\right\} > \\
&> \left\{\frac{1}{2}\int_0^1\left\{\left(\frac{d}{dx}(\theta v_1 + (1-\theta)v_2)\right)^2 + \alpha(\theta v_1 + (1-\theta)v_2)^2\right\}dx - \int_0^1 \beta^i(\theta v_1 + (1-\theta)v_2)dx + \\
&+ \frac{1}{2}\gamma_L\left[((\theta v_1 + (1-\theta)v_2) - u_L)^2\right]_{x=0} + \frac{1}{2}\gamma_R\left[((\theta v_1 + (1-\theta)v_2) - u_R)^2\right]_{x=1}\right\}
\end{aligned}
\tag{A2}
$$

Since

$$\theta\int_0^1 \beta^i v_1 dx + (1-\theta)\int_0^1 \beta^i v_2 dx = \int_0^1 \beta^i(\theta v_1 + (1-\theta)v_2)dx, \tag{A3}$$

it suffices to show that

$$
\begin{aligned}
&\theta\left\{\frac{1}{2}\int_0^1\left\{\left(\frac{dv_1}{dx}\right)^2 + \alpha v_1^2\right\}dx + \frac{1}{2}\gamma_L\left[(v_1-u_L)^2\right]_{x=0} + \frac{1}{2}\gamma_R\left[(v_1-u_R)^2\right]_{x=1}\right\} + \\
&+(1-\theta)\left\{\frac{1}{2}\int_0^1\left\{\left(\frac{dv_2}{dx}\right)^2 + \alpha v_2^2\right\}dx + \frac{1}{2}\gamma_L\left[(v_2-u_L)^2\right]_{x=0} + \frac{1}{2}\gamma_R\left[(v_2-u_R)^2\right]_{x=1}\right\} > \\
&> \left\{\frac{1}{2}\int_0^1\left\{\left(\frac{d}{dx}(\theta v_1 + (1-\theta)v_2)\right)^2 + \alpha(\theta v_1 + (1-\theta)v_2)^2\right\}dx + \\
&+ \frac{1}{2}\gamma_L\left[((\theta v_1 + (1-\theta)v_2) - u_L)^2\right]_{x=0} + \frac{1}{2}\gamma_R\left[((\theta v_1 + (1-\theta)v_2) - u_R)^2\right]_{x=1}\right\}
\end{aligned}
\tag{A4}
$$

To demonstrate the above inequality, it is enough to prove that

$$
\begin{aligned}
\theta\left(\frac{dv_1}{dx}\right)^2 + (1-\theta)\left(\frac{dv_2}{dx}\right)^2 &\geq \left(\frac{d}{dx}(\theta v_1 + (1-\theta)v_2)\right)^2 \\
\theta v_1^2 + (1-\theta)v_2^2 &> (\theta v_1 + (1-\theta)v_2)^2 \\
\theta(v_1-u_L)^2 + (1-\theta)(v_2-u_L)^2 &> ((\theta v_1 + (1-\theta)v_2) - u_L)^2 \\
\theta(v_1-u_R)^2 + (1-\theta)(v_2-u_R)^2 &> ((\theta v_1 + (1-\theta)v_2) - u_R)^2
\end{aligned}
\tag{A5}
$$

Since, for $\theta \in (0,1)$,

$$
\begin{aligned}
\theta a^2 + (1-\theta)b^2 - (\theta a + (1-\theta)b)^2 &= \\
= \theta a^2 + (1-\theta)b^2 - (\theta a)^2 - ((1-\theta)b)^2 - 2\theta(1-\theta)ab &= \\
= \theta(1-\theta)a^2 + \theta(1-\theta)b^2 - 2\theta(1-\theta)ab = \theta(1-\theta)(a-b)^2 &\geq 0
\end{aligned}
\tag{A6}
$$

we have that

$$
\theta\left(\frac{dv_1}{dx}\right)^2 + (1-\theta)\left(\frac{dv_2}{dx}\right)^2 \geq \left(\frac{d}{dx}(\theta v_1 + (1-\theta)v_2)\right)^2 \quad \Leftrightarrow \quad \theta(1-\theta)\left(\frac{dv_1}{dx} - \frac{dv_2}{dx}\right)^2 \geq 0
$$

$$
\theta(1-\theta)(v_1-v_2)^2 > 0 \quad \Leftrightarrow \quad
\begin{cases}
\theta v_1^2 + (1-\theta)v_2^2 > (\theta v_1 + (1-\theta)v_2)^2 \\
\theta(v_1-u_L)^2 + (1-\theta)(v_2-u_L)^2 > ((\theta v_1 + (1-\theta)v_2) - u_L)^2 \\
\theta(v_1-u_R)^2 + (1-\theta)(v_2-u_R)^2 > ((\theta v_1 + (1-\theta)v_2) - u_R)^2
\end{cases}
\tag{A7}
$$

Therefore, (A5) holds for any $\theta \in (0,1)$ and $v_1 \neq v_2$, ensuring the functional convexity.

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
