# Peer review of "A Procedure for Constructing the Solution of a Nonlinear Fredholm Integro-Differential Equation of Second Order"

_axioms, doi:10.3390/axioms11120672_

Round 1

Reviewer 1 Report

The paper is interested on “A procedure for constructing the solution of a nonlinear  Fredholm integro-differential equation of second order”. The paper presented a procedure for constructing its solution in a very simple and reliable way in which the only needed tool is the same one employed for solving a linear second order ordinary differential equation, subjected to Robin boundary conditions.  Also, the authors have presented some especial cases are simulated in order to illustrate the proposed tools.

The obtained theoretical results are good in this paper. The level of contribution is considered very well. However, there are some points that need to be addressed in this paper before considering it for acceptance in this journal. Therefore, a minor revision is needed by the authors according to the following suggestions:  

1.     There are some minor typos and grammatical errors in some parts of the text. Please double check all sentences and make sure that all equations are correct.

2.     The introduction needs to be improved by recent references.

3.     Can this work  be extended  using fractional operators? Such as Caputo-Hadamard fractional derivative ( see 10.1216/rmj.2021.51.17).

4.     In the conclusion section. Please make sure to suggest some possible specific future works based on the obtained results in this work such as generalizations or extensions or any valuable suggestions in one to two sentences only, so readers and other interested researchers can prepare a future plan for further research works concerning this particular study. For this, I suggest that the authors use some of these papers: 

https://doi.org/10.1002/mma.8039;  https://doi.org/10.3390/fractalfract6060289

Author Response

Reponses to Reviewer 1

Journal Axioms (ISSN 2075-1680)

Manuscript ID axioms-2017852

Comments and Suggestions for Authors

  1. There are some minor typos and grammatical errors in some parts of the text. Please double check all sentences and make sure that all equations are correct.

RESPONSE ## A paid editing service (from MDPI) was used in order to improve the text and avoid grammatical errors.

  1. The introduction needs to be improved by recent references.

RESPONSE ## As it can be seen, the introduction was improved and new references were added (also in the conclusions).

  1. Can this work  be extended  using fractional operators? Such as Caputo-Hadamard fractional derivative ( see 10.1216/rmj.2021.51.17).

RESPONSE ## I do not know. But I have inserted this reference…

  1. In the conclusion section. Please make sure to suggest some possible specific future works based on the obtained results in this work such as generalizations or extensions or any valuable suggestions in one to two sentences only, so readers and other interested researchers can prepare a future plan for further research works concerning this particular study. For this, I suggest that the authors use some of these papers: 

RESPONSE ##All the suggested papers were added to the references. Possible future works were suggested ( https://doi.org/10.1002/mma.8039;  https://doi.org/10.3390/fractalfract6060289)

Reviewer 2 Report

Comment for the article “Title

A PROCEDURE FOR CONSTRUCTING THE SOLUTION OF A NONLINEAR FREDHOLM INTEGRO-DIFFERENTIAL EQUATION OF SECOND ORDER”

Journal: Axioms

Manuscript ID: axioms-2017852

Authors: R. M. S. Gama , R. P. S. Gama

In this article,  a large class of nonlinear integro-differential equations, arising from the description of heat transfer problems, is considered. It is proposed a procedure for constructing its solution, where the needed tool is the same one employed for solving a linear second order ordinary differential equation, subjected to Robin boundary conditions. Proofs of convergence, existence and uniqueness are presented. In particular case, examples are given for illustrations.

Indeed, here ODEs of  (1) is considered and  the main aim of this article is to  generalize the mathematical description of cylindrical fins [15-20].  The topic is interesting and  similarity s fine,17%. See, the attached file on PDF.

Suggestion

-There is no information about the existence and uniqueness of solutions of ODEs in (1). It can be given.

-It is not clear how “equation (1) generalizes the mathematical description of cylindrical fins [15-20].”  Specific and satisfactory information can be given.

-The novelty and contribution of the article can be explained more clearly.

- There are some numerous punctuations and grammatical errors throughout the article. The article can be checked carefully. Necessary corrections can be done.

For example, “figure 1” should be “Figure  1”,

-The meaning of Table 1 can be explained clearly.

- The following papers  can be added to the references of this paper to rich them:

Continuability and boundedness of multi-delay functional integro-differential equations of the second order. Rev. R. Acad. Cienc. Exactas Fís. Nat. Ser. A Math. RACSAM 109 (2015), no. 1, 169–173.

A note on the stability and boundedness of solutions to non-linear differential systems of second order. Journal of the Association of Arab Universities for Basic and Applied Sciences 24  (2017), 169-175.

I would like to suggest  the acceptation of this article  in “Axioms” if the suggestions can be completed.

Author Response

Journal Axioms (ISSN 2075-1680)

Manuscript ID axioms-2017852

Suggestion

-There is no information about the existence and uniqueness of solutions of ODEs in (1). It can be given.

RESPONSE ### In fact the uniqueness are shown in section 8. The existence is ensured from the existence of the limit of the sequence.

-It is not clear how “equation (1) generalizes the mathematical description of cylindrical fins [15-20].”  Specific and satisfactory information can be given.

RESPONSE ### In references 19 and 20 we have some particular models. Equation (1) generalizes the description because it allows the direct thermal radiant interchange among far positioned points (pleas see reference 24).

 -The novelty and contribution of the article can be explained more clearly.

RESPONSE ### Theses issues were added in the conclusion.

 - There are some numerous punctuations and grammatical errors throughout the article. The article can be checked carefully. Necessary corrections can be done.

RESPONSE ### A paid editing service (from MDPI) was used in order to improve the text and avoid grammatical errors. Please find enclosed the "certificate".

For example, “figure 1” should be “Figure  1”,

RESPONSE ### Corrected in the revised version.

 -The meaning of Table 1 can be explained clearly.

RESPONSE ### An explanation was inserted. 

- The following papers  can be added to the references of this paper to rich them:

RESPONSE ### Thank you for these suggestions. All the suggested references were added.

 … Continuability and boundedness of multi-delay functional integro-differential equations of the second order. Rev. R. Acad. Cienc. Exactas Fís. Nat. Ser. A Math. RACSAM 109 (2015), no. 1, 169–173.

 … A note on the stability and boundedness of solutions to non-linear differential systems of second order. Journal of the Association of Arab Universities for Basic and Applied Sciences 24  (2017), 169-175.

 I would like to suggest  the acceptation of this article  in “Axioms” if the suggestions can be completed.

Reviewer 3 Report

The paper is potentially interesting but serious flaws have to be removed after reconsider it:

Major points

1. the general structure of the paper is very poor: the paragraph are not numbered and the introduction is too powerless for a potential reader. The conclusions too. In the revision these parts must be strongly reinforced 

2. in the variational part are the authors really sure that the functional is convex? The simple statement which refers to two paper 27 and 28 (see references) are not enough. This aspect must be clarify in a right way

Minor points

1. Some paper should be quoted and discussed in reinforcing the present level of elaboration of the paper (the first two below) and the other one for possible new scenarios of development:

- M. Rabbani, B. Zarali, Solution of Fredholm Integro-differential Equations System by Modified Decomposition Method, Journal of Mathematics and Computer Science, 5 (2012), no. 4, 258 - 26

- Ferrara M, Bianca C, Guerrini L., Asymptotic limit of an integro-differential equation modelling complex systems, "Izvestiya. mathematics", Articolo in rivista, n. 78, 2014, ISSN: 1064-5632.

Recommendation:

I suggest a deep revision after reconsidering the paper 

  1.  
  2.  

Author Response

Journal Axioms (ISSN 2075-1680)

Manuscript ID axioms-2017852

Comments and Suggestions for Authors

The paper is potentially interesting but serious flaws have to be removed after reconsider it:

Major points

  1. the general structure of the paper is very poor: the paragraph are not numbered and the introduction is too powerless for a potential reader. The conclusions too. In the revision these parts must be strongly reinforced 

RESPONSE #### The paragraphs are now numbered. The introduction as well as the conclusions were improved.

  1. in the variational part are the authors really sure that the functional is convex? The simple statement which refers to two paper 27 and 28 (see references) are not enough. This aspect must be clarify in a right way

RESPONSE #### In order to satisfy this request, an Appendix was inserted (proving the functional's convexity)

Minor points

  1. Some paper should be quoted and discussed in reinforcing the present level of elaboration of the paper (the first two below) and the other one for possible new scenarios of development:

RESPONSE #### All the suggested papers (below) were added in the references.

- M. Rabbani, B. Zarali, Solution of Fredholm Integro-differential Equations System by Modified Decomposition Method, Journal of Mathematics and Computer Science, 5 (2012), no. 4, 258 - 26

- Ferrara M, Bianca C, Guerrini L., Asymptotic limit of an integro-differential equation modelling complex systems, "Izvestiya. mathematics", Articolo in rivista, n. 78, 2014, ISSN: 1064-5632.

  1. - Ferrara M, Munteanu F, Udriste C, Zugravescu D, Controllability of a nonholonomic macroeconomic system, "Journal of optimization theory and applications",   154, 2012, pp. 1036-1054, ISSN: 0022-3239.

Recommendation:

I suggest a deep revision after reconsidering the paper 

Round 2

Reviewer 2 Report

Comment for the REVISED article “Title

A PROCEDURE FOR CONSTRUCTING THE SOLUTION OF A NONLINEAR FREDHOLM INTEGRO-DIFFERENTIAL EQUATION OF SECOND ORDER”

Journal: Axioms

Manuscript ID: axioms-2017852

Authors: R. M. S. Gama , R. P. S. Gama

Suggestions and Answers

-There is no information about the existence and uniqueness of solutions of ODEs in (1). It can be given.

Answer: Suitable and new information have been given for ODEs (1) in (1).

-It is not clear how “equation (1) generalizes the mathematical description of cylindrical fins [15-20].”  Specific and satisfactory information can be given.

Answer: Reply to this comment is given as” In references 19 and 20 we have some particular models. Equation (1) generalizes the description because it allows the direct thermal radiant interchange among far positioned points (please see reference 24)”.

 -The novelty and contribution of the article can be explained more clearly.

-The novelty and contribution of the article can be explained more clearly.

Answer: In “ 12. Conclusion”, new information have been added about these  comments.

- There are some numerous punctuations and grammatical errors throughout the article. The article can be checked carefully. Necessary corrections can be done.

For example, “figure 1” should be “Figure  1”,

-The meaning of Table 1 can be explained clearly.

Answer. Some  suitable corrections have been done.

- The following papers can be added to the references of this paper to rich them.

Answer: It was done accordingly.

I would like to suggest the acceptation of the REVISED  article  in “Axioms”.

Reviewer 3 Report

I have read the paper after changes and I consider this last final version fully acceptable for publication. I recommend the acceptation in its final version